# Knee-High Devices Are Gold in Closing the Foot Ulcer Gap: A Review of Offloading Treatments to Heal Diabetic Foot Ulcers

**DOI:** 10.3390/medicina57090941

**Published:** 2021-09-06

**Authors:** Peter A. Lazzarini, Gustav Jarl

**Affiliations:** 1School of Public Health and Social Work, Queensland University of Technology, Brisbane 4059, Australia; 2Allied Health Research Collaborative, The Prince Charles Hospital, Brisbane 4032, Australia; 3Department of Prosthetics and Orthotics, Faculty of Medicine and Health, Örebro University, SE-70182 Örebro, Sweden; gustav.jarl@regionorebrolan.se; 4University Health Care Research Center, Faculty of Medicine and Health, Örebro University, SE-70182 Örebro, Sweden

**Keywords:** activity, adherence, diabetic foot, diabetic foot ulcer, offloading devices, offloading treatment, peripheral neuropathy, plantar pressure, plantar tissue stress, pressure offloading

## Abstract

Diabetic foot ulcers (DFU) are a leading cause of the global disease burden. Most DFUs are caused, and prolonged, by high plantar tissue stress under the insensate foot of a person with peripheral neuropathy. Multiple different offloading treatments have been used to try to reduce high plantar tissue stress and heal DFUs, including bedrest, casting, offloading devices, footwear, and surgical procedures. The best offloading treatments are those that balance the benefits of maximizing reductions in high plantar tissue stress, whilst reducing the risks of poor satisfaction, high costs and potential adverse events outcomes. This review aimed to summarize the best available evidence on the effects of offloading treatments to heal people with DFUs, plus review their use in clinical practice, the common barriers and solutions to using these treatments, and discuss promising emerging solutions that may improve offloading treatments in future. Findings demonstrate that knee-high offloading devices, non-removable or removable knee-high devices worn for all weight-bearing activities, are the gold standard offloading treatments to heal most patients with DFU, as they are much more effective, and typically safer, quicker, and cheaper to use compared with other offloading treatments. The effectiveness of offloading treatments also seems to increase when increased offloading mechanical features are incorporated within treatments, including customized insoles, rocker-bottom soles, controlled ankle motion, and higher cast walls. However, in clinical practice these gold standard knee-high offloading devices have low rates of prescription by clinicians and low rates of acceptance or adherence by patients. The common barriers resulting in this low use seem to surround historical misperceptions that are mostly dispelled by contemporary evidence. Further, research is now urgently required to close the implementation gap between the high-quality of supporting evidence and the low use of knee-high devices in clinical practice to reduce the high global disease burden of DFU in future.

## 1. Introduction

Diabetic foot ulcers (DFUs) are a leading cause of global hospitalization, amputation, and disability [1,2,3,4,5]. A DFU is a full-thickness wound on the foot of a person with diabetes, usually accompanied by diabetic peripheral neuropathy (DPN) and/or peripheral artery disease (PAD) [6,7]. Each year, DFUs affect ~20 million people globally with another ~130 million having DPN placing them at risk of a DFU [1,4,5]. People with DFUs typically take months to heal, are intensive to treat, are at high risk of hospitalization and amputation, and, even once healed, DFUs are highly recurrent [1,2,5,8]. 

The most common way to develop a DFU is by high plantar tissue stress on the insensate foot of a person with DPN [1,9,10,11]. Plantar tissue stress is a product of plantar pressure and weight-bearing activity over time [9,10,11]. In people without DPN, areas of high plantar tissue stress typically lead to pain, which usually results in the person changing their gait or seeking treatment to “offload” the painful area, such as via a limp or footwear [9]. Yet, in people with DPN, the sensory neuropathy results in a loss of sensation to pain, which usually means the person does not change their gait or seek treatment to offload [1,9,10,11]. The motor neuropathy from DPN can also lead to further detrimental gait, balance, and bony deformity changes, which can lead to even higher plantar tissue stress [1,9,10,11,12,13,14]. High plantar tissue stress, left untreated, often results in a breakdown of subcutaneous plantar tissue and a DFU develops, and if it remains untreated considerably prolongs DFU healing [1,9,10,11]. Thus, offloading treatment to reduce the high plantar tissue stress that caused the DFU is critical to heal the DFU [1,9,10,11]. 

The main aims of offloading treatments are to maximize the desirable effects (or benefits) of reducing high plantar tissue stress, such as reducing high plantar pressure, high weight-bearing activity, and increasing treatment adherence; whilst minimizing any undesirable effects (or risks), such as adverse events, poor patient satisfaction, and high costs [7,9,10,11]. Different offloading treatments have been used over the years in clinical practice to try to address these aims and more effectively heal DFUs, such as via casting, offloading devices, footwear, and surgical procedures [9,10,11]. Yet, studies have found these different offloading treatments offer different levels of desirable and undesirable effects and in turn produce different healing outcomes [9,10,11]. Additionally, studies have indicated that best practice offloading treatments have low rates of use; i.e., low prescription by clinicians [8,15,16,17] and low adherence by patients [18,19,20,21,22,23]. Further studies suggest there are many potential barriers and solutions to the use of these best practice offloading treatments [15,20,21,24]. Thus, the aims of this paper were to review the best available evidence firstly on the effects of offloading treatments to heal people with DFU, secondly on the use of offloading treatments in clinical practice, thirdly on the common barriers and solutions to using these offloading treatments, and lastly discuss promising emerging solutions that may improve offloading treatments in future. 

## 2. Evidence on the Effects of Offloading Treatments

The concept of foot ulcers being caused by high plantar pressure on insensate feet has been around since the 1950s following the early leprosy research by Dr Paul Brand [25,26,27]. In the 1980s, the concept was confirmed for DFUs following the early diabetes research by Professor Andrew Boulton [25,28,29,30], and in the 2000s, the concept was expanded to include high weight-bearing activity over time by Professor Michael Mueller, and called high plantar tissue stress [31,32]. Following knowledge of these concepts, most considered near complete offloading of all plantar pressure was the best aim for offloading treatment and used treatments such as bedrest, crutches, wheelchairs, and casting [10,11]. Yet, early studies in the 1990s, by Professors Mueller, Armstrong, Lavery, and others, revealed complete offloading to not be as effective as first thought, and instead total contact casts (TCC) were found to be the gold standard in offloading treatment to heal DFUs [33,34,35]. 

Over 160 studies (including ~40 controlled trials) have now investigated the effects that different offloading treatments have on healing people with DFU [10]. Fortunately, systematic reviews [10,36,37,38,39,40,41,42] and evidence-based guidelines [11,43,44,45] have systematically weighed up all the available evidence on the effects of offloading treatments from these studies to develop best practice offloading treatment recommendations to heal DFU [11,43,44,45]. In Table 1, to try and simplify this large volume of complex offloading evidence for the lay clinician [10,11,36,37,38,39,40,41,42,43,44,45], we have summarized the collective findings on the effects that different offloading treatments have on the most important outcomes for patients and clinicians when deciding on offloading treatments to heal DFUs [6,7,11], i.e.,
○Healing, such as proportions of DFU healed;○Plantar pressure, such as peak plantar pressure on the DFU site;○Weight-bearing activity, such as daily steps; ○Adherence to treatment, such as proportion of daily steps using the treatment;○Patient satisfaction with treatment, such as impact on daily activities;○Cost-effectiveness, such as healing outcomes in relation to all treatment costs; ○Adverse events, such as new ulcers, falls and amputations; ○Contraindications for treatment, such as infection, ischemia or falls risk; and○Patient intolerance factors to treatment, such as increased driving requirements. 

In the remainder of this section, we will further attempt to more simply explain the evidence-based effects that different offloading treatments have on these important outcomes and how best practice guidelines developed their first- (gold standard) to last-choice (last resort) recommendations for offloading treatments to heal DFU [11,43,44,45].

### 2.1. Non-Removable Knee-High Devices: First-Choice Offloading Treatment (Gold Standard)

The first-choice (or gold standard) offloading treatment recommended by all guidelines to most effectively heal a plantar DFU are non-removable knee-high offloading devices (Table 1) [11,43,44]. This recommendation is based upon five meta-analyses [36,37,38,39,40], of 14 trials that included a total of 758 participants [33,34,35,46,47,48,49,50,51,52,53,54,55,56], showing that patients treated with non-removable knee-high devices have moderately better desirable effects on healing compared to those treated with removable knee-high devices [10,36,37,38,39,40]. This means, according to best evidence, moderately more patients are likely to completely heal their ulcer, and quicker, when prescribed non-removable knee-high devices than if prescribed removable knee-high devices [10,36,37,38,39,40]. Unsurprisingly, this moderately better healing outcome is the result of moderately better desirable effects on adherence in those using non-removable knee-high devices [10], i.e., patients prescribed non-removable knee-high devices are likely to wear them for moderately longer then removable knee-high devices [18,19,20] and in turn get moderately better healing [10]. Otherwise, non-removable and removable knee-high devices had similar effects on nearly all other important outcomes, except for cost-effectiveness which was also better with non-removable devices [10,36]. Again, this is unsurprising considering these devices are the same, except for the impact of removability on adherence [10,11]. Further, best evidence suggests non-removable knee-high devices have even larger desirable effects on healing, plantar pressure, weight-bearing activity, adherence, and cost-effectiveness when compared to all other offloading treatments [10,36,37,38,39,40,41,42]. Thus, non-removable knee-high devices are strongly recommended as the first-choice offloading treatment to heal plantar DFU [11,43,44].

A non-removable knee-high device is defined as an offloading device that extends up the leg to just below the knee and cannot be removed by the patient, including customized TCCs and prefabricated removable knee-high cast walkers made non-removable using an irremovable wrap (known as “instant TCCs” (iTCCs)) [11,43,44]. Non-removable knee-high devices typically contain all the offloading mechanical features that are thought important to reduce the most amount of high plantar tissue stress [10,11]. Figure 1 illustrates the increasing effects on reducing high plantar tissue stress that increasing offloading mechanical features have when incorporated in different offloading treatments, such as customized insoles/foot-device interfaces, rocker-bottom soles, controlled ankle motion, higher cast walls/struts, and being non-removable [10,11,57,58,59,60,61]. Trials show iTCCs and TCCs have similar effects on healing and other outcomes, except for small desirable effects in favor of iTCCs for better patient satisfaction and cost-effectiveness [10,36,40,49,55,62,63]. Thus, with similar mechanical features producing similar outcomes, all guidelines now recommend that either TCCs or iTCCs should be prescribed as the first-choice offloading treatment to heal DFU as long as the patient is not contraindicated, can tolerate wearing the device and their foot fits safely in the device (i.e., unlikely to cause adverse events) [11,43,44]. Contraindications to prescribing either type of non-removable knee-high device are now considered to include moderate-to-severe ischemia, moderate-to-severe infection and having a high falls risk [11,43,44]. Factors likely to impact on patient intolerance to using these devices include certain job requirements, frequent need to drive, infrequent access to DFU care, and hot climates [11,43,44]. 

The figure is not to scale and is designed to illustrate the increasing capacity to reduce high plantar tissue stress with the incorporation of additional mechanical features in different offloading treatments. Colored bars represent the mechanical features usually incorporated in different offloading treatments: Gold = first-choice (gold standard); silver = second-choice; bronze = third-choice; orange= last-choice; red = no-choice.

### 2.2. Removable Knee-High Devices: Second-Choice Offloading Treatment

In those DFU patients where non-removable knee-high devices are contraindicated or not tolerated, then the second-choice offloading treatment recommended by guidelines are removable knee-high offloading devices (Table 1) [11,43,44]. Interestingly, this recommendation is based upon trials showing that removable knee-high and removable ankle-high devices have similar effects on healing outcomes [10,36,64]. However, removable knee-high devices have small desirable effects on reducing more high plantar pressure and activity, but small undesirable effects on adherence compared to removable ankle-high devices [10,36,64]. This means, according to best evidence, patients prescribed a removable knee-high device compared to being prescribed a removable ankle-high device are likely to reduce slightly more high plantar pressure [58,61,64,65,66,67,68], slightly more high weight-bearing activity [23,34,54,64], but adhere to using their device for slightly less time [18,19,20,61,64]; and these effects balance each other out to give similar healing outcomes [10,36,64]. However, this also suggests if patients adhere to using these devices for the same amount of time, then this should result in more high plantar tissue stress being reduced (plantar pressure and activity) and in turn a better healing outcome in those prescribed the knee-high option [10,11,44]. Otherwise, available evidence suggests that patient satisfaction, cost-effectiveness, and adverse events are similar for both removable devices [10,11,44]. Thus, removable knee-high devices are weakly recommended as the second-choice offloading treatment, when non-removable knee-high devices are contraindicated or not tolerated [11,44].

A removable knee-high offloading device is an offloading device that extends up the leg to just below the knee and is able to be removed by the patient [11,44]. These devices mainly include prefabricated removable knee-high cast walkers (RCWs) and customized bivalved TCCs [11,44]. Removable knee-high devices typically contain all the same offloading mechanical features as non-removable knee-high devices, except for non-removability (Figure 1) [11,44]. Thus, with removable knee-high devices containing the same mechanical features as non-removable knee-high devices, it is thought that if patients adhered to using them for all (100%) of their weight-bearing activity, as enforced by non-removable devices, then they should also get the same gold standard healing outcome [10,11,44]. This suggests that as long as patients adhere to using their removable knee-high device at all times they are on their feet, then they should get near gold standard treatment, even if having to remove their device occasionally during activities that involve no or a minimum of weight-bearing, such as driving or bathing [10,11,44]. However, this is yet to be consistently confirmed in trials [55,56], and until then removable knee-high offloading devices, if tolerated and not contraindicated, are considered the second-choice offloading treatment along with encouraging the patient to wear the devices for all of their weight-bearing activities [10,11,44]. Contraindications for removable knee-high devices are fewer, but still include severe ischemia, severe infection, and high falls risk, with intolerance factors to consider including certain job requirements and hot climates [10,11,44].

### 2.3. Removable Ankle-High Devices: Third-Choice Offloading Treatment

In those patients where knee-high devices are contraindicated or not tolerated, then the third-choice offloading treatment recommended by guidelines are removable ankle-high offloading devices (Table 1) [11,44]. As mentioned, removable ankle-high devices have been found in trials to give similar healing outcomes to removable knee-high devices, mainly due to the increased adherence found in those wearing ankle-high devices [10,36,64]. Furthermore, best available evidence suggests removable ankle-high devices compared with using therapeutic footwear have moderately better healing outcomes, made up of the small desirable effects on reducing both high plantar pressure [58,61,65,67,68,69,70,71,72] and weight-bearing activity, and similar adherence effects [10,36,64]. This means, patients prescribed removable ankle-high devices are likely to get similar healing outcomes to removable knee-high devices because they are likely to adhere to using them longer, and moderately better healing outcomes than therapeutic footwear because they are likely to reduce more high plantar pressure, more weight-bearing activity and have similar adherence levels [10,11,44]. Otherwise, available evidence suggests patient satisfaction and adverse events are similar, but removable ankle-high devices are moderately more cost-effective than therapeutic footwear [10,36,64]. Thus, removable ankle-high devices are strongly recommended as the third-choice offloading treatment, when knee-high devices are contraindicated or not tolerated [11,44].

A removable ankle-high offloading device is defined as an offloading device that extends no higher up the leg than just above the ankle and may contain offloading mechanical features such as low cast walls, controlled ankle motion, rocker-bottom sole, and customized insole/foot-device interface [11,44]. Thus, this category of device includes a large heterogenous range of devices, including ankle-high walkers, forefoot offloading shoes, rearfoot offloading shoes, cast slippers/shoes, and postoperative healing shoes [11,44]. The main offloading mechanical feature that is different between the removable knee-high and ankle-high devices is the height of the cast walls (or struts) [11,44]. Studies have found that the higher the cast walls in the device, the higher the load taken up by the cast walls and in turn the more plantar pressure is redistributed away from the plantar surface of the foot, and the more high plantar pressure reduced [58,60,61]. Yet, other important offloading mechanical features differ across different removable ankle-high devices with some including all other mechanical features, such as ankle-high walkers, and others no mechanical features, such as flat post-operative healing shoes [10,11,44]. Thus, guidelines suggest in future studies that removable ankle-high devices should be further subgrouped based on their mechanical features, such as above ankle-high and below ankle-high devices [11,44], or other sub-groupings. Regardless, most of these ankle-high devices have very few contraindications and should be tolerated by most patients [10,11,44]. Thus, if knee-high devices are contraindicated or not tolerated, then a removable ankle-high device should be prescribed, with preference for those with more beneficial mechanical features, such as ankle-high walkers [11,44]. Regardless, the patient should be encouraged to adhere to wearing the device for as much of their weight-bearing activity as possible, as the more they wear their device the quicker their ulcer is likely to heal [11,44].

### 2.4. Footwear: Last-Choice Offloading Treatment

In those patients unable to tolerate or access any offloading devices, then the last-choice offloading treatment recommended by some guidelines is (therapeutic) footwear (Table 1) [11,44,45]. The specific recommendation is different in different guidelines with some recommending therapeutic footwear only and others recommending appropriately fitting (conventional or therapeutic) footwear with felted foam padding [11,44,45]. Regardless, all recommend using footwear as a last resort when no offloading devices are available, and not to use conventional footwear only [11,44,45]. This is because trials show that therapeutic footwear, compared with all offloading devices, have small-to-large undesirable effects on healing, plantar pressure reduction, weight-bearing activity, and cost-effectiveness [10,11,36,37,40,44]. This means patients prescribed therapeutic footwear are likely to have much worse healing outcomes, because they will get much less reduction of high plantar pressure, do much more weight-bearing activity, but adhere to wearing the footwear slightly more than if getting any offloading device [10,11,36,37,40,44]. Furthermore, whilst conventional footwear has not been tested against offloading devices for healing outcomes, it has been shown to have less effect on reducing high plantar pressure than therapeutic footwear [73,74,75,76,77,78,79,80,81,82,83] and much less than offloading devices [58,61,65,67,68,69,70,71,72], and thus, thought to be a much worse offloading treatment option than therapeutic footwear and hence advised against [10,36,44,45]. Felted foam on the other hand, although not trialed in footwear for healing [10], has been trialed in removable ankle-high devices where it was found to produce small desirable effects on healing outcomes [84,85,86] and reducing high plantar pressure [87,88] compared with the same removable ankle-high devices alone [10,11,44]. Therefore, it is thought that adding felted foam redistributive padding to appropriately fitting footwear should also help produce a slightly better effect on healing than footwear alone [11,44]. Thus, therapeutic footwear or appropriately fitting conventional footwear with felted foam redistributive padding is weakly recommended when no other offloading devices are available, plus, it is strongly recommended not to use conventional footwear alone as an offloading treatment to heal a DFU [11,44,45]. 

Therapeutic footwear is defined as footwear designed to have a therapeutic effect that cannot be provided by conventional footwear and typically contains the offloading mechanical features of customized insoles and/or rocker-bottom soles [11,44,45], such as custom-made footwear, customized footwear, and prefabricated extra-depth footwear [11,44,45]. Conventional footwear is defined as off-the-shelf footwear with no specific intended therapeutic effect and no specific offloading mechanical features, such as walking shoes, running shoes, oxford shoes, sandals, and slippers [11,44,45,89,90]. Felted foam is typically made from either a combined felt and foam material, or from felt alone, that has different densities and an adherent backing that enables it to be cut, contoured, and fixed to the foot or the insole of an offloading device or footwear [11,44,45]. Again, these options should be a last resort as nearly all options will lack most offloading mechanical features that typically result in significant reductions in high plantar pressure [10,11,44,45]. Thus, routinely using these last resort options in patients with DFU is likely to result in patients with much slower healing or ulcers not healing, potentially more adverse events due to this prolonged healing (such as infections, hospitalizations and amputations) and overall much higher treatment costs to the patient and service, than when routinely using offloading devices [10,11,44]. Thus, therapeutic footwear or appropriately fitting conventional footwear with redistribute felted foam padding should always be a last-choice offloading treatment [11,44,45].

### 2.5. Surgical Offloading Procedures: Non-Healing Ulcer Offloading Treatment

In those patients that are non-healing, after using the best offloading treatment indicated and tolerable for them, then they should be considered for surgical offloading procedures (Table 1) [11,44]. A non-healing ulcer is typically defined as a DFU that has not reduced in size by >50% within 4–6 weeks of best practice DFU treatment [7,11,44]. Best practice DFU treatment importantly is usually defined as receiving evidence-based DFU classification assessment, local wound debridement, appropriate wound dressings, antibiotics (if infected), revascularization (if ischemic), along with the above best offloading treatment [7,11,44]. The reason that surgical offloading procedures are recommended for non-healing ulcers is that most trials of these procedures were in those with non-healing ulcers [10,11,41,44]. These trials found that various surgical offloading procedures (when used in conjunction with knee-high offloading devices) mostly produce small-to-moderate desirable effects on better healing [91,92,93,94,95,96,97,98,99,100,101], plantar pressure reductions [102], and longer-term patient satisfaction [95,103], but they also tend to have small undesirable effects on higher adverse events and higher costs [91,92,93,95,99] compared to using knee-high offloading devices alone in patients with non-healing DFU [10,11,41,44]. Thus, surgical procedures are weakly recommended in those patients that have attempted best practice offloading treatments in conjunction with all other best practice DFU treatments for ~6 weeks and this has not helped heal their ulcers [11,44].

Surgical offloading procedures are defined as a surgical procedure undertaken with the intention of relieving mechanical stress (pressure) from a specific region of the foot [10,11,41,44]. For non-healing forefoot ulcers this has been evidenced to include metatarsal head resection, joint arthroplasty, osteotomy, Achilles’ tendon lengthening, or gastrocnemius recession [10,11,41,44,100,101]. Whereas, for non-healing plantar digital ulcers this has been evidenced to include digital flexor tenotomies [10,11,44,100,101]. Contraindications to these surgical procedures include moderate-to-severe ischemia, moderate-to-severe infection, and moderate-to-severe edema [10,11,44]. Intolerance factors to consider include job requirements, frequent driving requirements, and infrequent access to ongoing rehabilitation care [11,44]. Thus, if best practice DFU treatment (including best offloading devices) reduces the ulcer in size by <50% after 4–6 weeks of this treatment, then it is weakly recommended to consider using surgical offloading procedures depending on the location of the ulcer and taking into account contraindications and patient tolerance [11,44]. 

### 2.6. Summary of Best Offloading Treatments

In summary, prescribing best practice offloading treatments should involve clinicians and patients discussing the best treatment that balances the benefits of maximizing reductions in high plantar tissue stress, whilst reducing the risks of poor satisfaction, high costs and potential adverse events for the patient. The treatments that typically best achieve this balance are ones that include as many of the important offloading mechanical features as possible (see Figure 1), that patients can adhere to using for all weight-bearing activity and that don’t cause more problems, such as new ulcers or falls. Thus, offloading treatments that include all those features are considered gold standard (i.e., knee-high offloading devices), and conversely those treatments including none of those features are recommended not to use (i.e., conventional footwear). Lastly, whilst most evidence for these offloading treatments are from trials of patients with plantar forefoot and midfoot DFUs, the limited evidence for plantar rearfoot DFUs also indicates similar offloading treatment choices should still produce the most benefit on healing rearfoot DFUs as well.

## 3. Common Barriers and Solutions to Using Best Offloading Treatments

### 3.1. Use in Clinical Practice

As outlined above, there is now high-quality evidence that non-removable knee-high devices, and even removable knee-high devices worn for ~100% of weight-bearing activity, are the gold standard of offloading treatments. Yet, observational studies have found these knee-high devices are somewhat rarely prescribed and used in clinical practice [15,16,17]. One common hypothesis for such a low prescription rate was that whilst demonstrated in the controlled environment of trials [10], many clinicians still considered knee-high devices would be much less effective in the real-world where multiple other factors also influence healing outcomes [10]. However, recent large real-world cohort studies seem to have dispelled this hypothesis [8,104]. For example, a retrospective study of ~12,000 US patients and a prospective study of ~5000 Australian patients found, after adjusting for up to 30 other factors including other common DFU treatments, that those prescribed knee-high devices were significantly more likely to heal by 3 and 12 months, with quicker healing times and less amputations, than those receiving all other offloading treatments [8,104].

Another common hypothesis was that at least half of all patients presenting with DFU would be contraindicated to using gold standard knee-high offloading device treatment because of ischemia or infection [10]. However, guidelines now suggest mild ischemia and mild infection are not contraindications for non-removable knee-high devices, and moderate ischemia and moderate infection are not contraindications for removable knee-high devices [11,44]. Thus, gold standard knee-high offloading treatment options are now really only contraindicated in those with severe ischemia or severe infection [10], which equates to ~10% of all DFU patients, with the exception of those with high falls risks [8,104,105]. Furthermore, available trial evidence suggests similar falls outcomes in those using different offloading devices [11,34,47,50,54,55,56], and emerging evidence suggests that those using knee-high devices in conjunction with contralateral shoe raises and other mobility aides, may in fact by more stable than without these gold standard devices [61,106,107]. Thus, this further suggest that only a small proportion of people with DFU are contraindicated to gold standard knee-high offloading treatment options [10,11,44].

These large cohort studies, along with multiple clinical practice surveys report though, that <40% of patients in the US, Europe and Australia are actually prescribed knee-high devices, with about half of those being removable [8,15,16,17,104,105]. Further, of those prescribed removable knee-high devices, other studies in the US, UK, and Middle East found patients only adhered to using them for ~30–60% of their weight-bearing activity [10,18,19,20]. Therefore, it seems whilst nearly all DFU patients are indicated for gold standard knee-high offloading treatment options, that at best anywhere in the world, far fewer than half of all patients are actually prescribed them, and of those, they only use them for around half of their necessary treatment times. This suggests that large gaps in best practice offloading treatments are occurring across the world and is likely resulting in highly preventable poor DFU outcomes for patients and highly preventable high disease burdens for nations. In the remainder of this section, we will further review the common barriers and solutions identified in the literature to using these gold standard offloading treatment options for clinicians and patients.

### 3.2. Barriers and Solutions to Clinicians Prescribing

The most commonly identified barriers for clinicians to prescribing these best practice offloading treatments, include the lack of necessary knowledge, expertise, time, and costs to apply these gold standard offloading treatments [10,11,20,24,44]. Firstly, whilst many clinicians now know that TCCs are gold standard, there still seems to be a general lack of knowledge that the more user-friendly “instant TCCs” (iTCCs), and removable cast walkers (RCWs) when adhered to at all times, are also gold standard [10,11,44]. Further, there seems to be a lack of knowledge about the critical impact that adherence to using these treatments has on healing outcomes, and that if removable knee-high devices are used for all weight-bearing activity they are likely to produce gold standard healing outcomes [9,10,11,44]. With only a small proportion of DFU patients contraindicated and intolerant to knee-high devices, this means that gold standard offloading treatment options are much more readily available for clinicians to prescribe to most patients [9,10,11,44]. Therefore, any improved dissemination of the knowledge that a range of knee-high devices can be considered gold standard for the vast majority of DFU patients, should also improve DFU treatment and healing outcomes [11,44].

Secondly, whilst a lack of expertise may have been the case when TCCs were the only gold standard offloading treatment option in older guidelines, this is not the case now iTCCs are also considered gold standard [10,11,36,44]. iTCCs are prefabricated RCWs made irremovable and many clinicians are now very familiar with RCWs as they are readily used for multiple other health conditions, such as fractures of the lower leg [11,44]. Thus, with the exception of the minority of DFU patients with large foot deformities or severe edema that will still need more complex TCC application, most clinicians should already be familiar with and have the minimal necessary skills required to apply these user-friendly prefabricated RCW or iTCC treatments [11,44].

Lastly, a lack of time and costs seem to also be a hangover perception from when TCCs were the only gold standard treatment option available [10,11,36,44]. Studies now show that applying an iTCC requires much less time and cost than a TCC, and also potentially less time and cost than most other offloading treatments, particularly as more contemporary RCWs often cost <$US50 [11,44]. Furthermore, recent high quality cost-effectiveness analyses show overall treatment costs using iTCCs are ~17–23% cheaper than using a TCC whilst producing similar healing outcomes, ~39–46% cheaper than a RCW whilst producing moderately better healing outcomes (due to lower adherence found to RCWs), and ~41–55% cheaper than therapeutic footwear whilst producing considerably better healing outcomes [10,36]. Thus, in summary, gold standard offloading treatment options should now require the same or less knowledge, expertise, time, and costs to apply than nearly all other offloading treatments, and thus readily accessible to prescribe for most clinicians to most of their DFU patients [11,44].

### 3.3. Barriers and Solutions to Patients Adhering

Once prescribed, the most commonly identified barriers in the literature for patients to adhere to using gold standard offloading treatment options include patient misperception, poor mobility, low motivation, and the higher weight of these treatments [10,18,19,20,23,24,108,109]. Firstly, studies indicate patients perceive that they only need to adhere to their offloading treatment for weight-bearing activities outside of their home [20,24]. This misperception is likely due to patients self-reporting that they do limited weight-bearing activity inside their home, plus, the perception that inside the home is considered safe and clean in many cultures [20,24,45,110]. Yet, unfortunately, studies show that DFU patients actually perform most of their weight-bearing activity inside the home [10,19,20,22,110]. Thus, it seems patients misperceive that they are adhering to their treatment for nearly all their (outside) weight-bearing activities, when in fact they are not adhering for nearly all of their (inside) weight-bearing activities [18,19,20]. Therefore, educating patients to understand that total adherence is actually defined as wearing their offloading devices for all activities in which they are on their feet, both inside and outside of their home, may improve patient adherence [9,20,24].

Secondly, poor mobility and stability when using these devices has been identified as barriers for patient adherence. Yet, studies have found that simple contralateral shoe raises to correct device-induced leg length discrepancy and instability, along with walking aides when required, have addressed many of these mobility barriers, improved stability, increased patient satisfaction and inturn adherence rates [11,44,61,106,107]. Thus, the prescription of offloading devices along with relatively inexpensive prefabricated contralateral shoe raises and mobility aides, such as wheely walkers or frames, may not only improve high plantar pressure reduction and healing outcomes, but also improve patient’s perception of stability and reduce falls [11,44].

Thirdly, low patient motivation to using offloading devices has also been found to be linked to those patients perceiving to have less severe DFU [18,20,24,108]. Studies on offloading adherence have found that DFU patients with larger ulcers, neuropathic pain or ischaemic pain were more likely to adhere to their devices then patients with smaller ulcers and no pain [18,20]. Thus, educating patients on the knowledge that all DFUs should be considered a severe condition, which they are unfortunately unable to feel due to their DPN, and that offloading devices are highly likely to reduce the severity, and heal their DFU if worn consistently, is likely to help improve adherence [20,111]. Studies suggest that patient-centric education, along with motivational interviewing techniques, also show promise in improving low patient motivation to offloading adherence. These studies recommend using layman’s terms, such as “fragile foot” or “broken foot”, to educate patients that DFUs are in fact a severe condition and that their offloading treatments need to be used to protect their “fragile foot” against what seems like “trivial trauma” (from plantar tissue stress that they cannot feel due to DPN) to prevent “further damage” and allow their “fragile foot” to rest and heal [24,108,111,112]. Additionally, visually showing patients their plantar pressure maps and healing rates, before and after offloading treatment is used, have also shown promise in improving patient’s understanding of the severity and impact of high plantar pressure on DFUs, and in turn adherence to offloading treatments [20,24,108,109,111,113].

Finally, studies have shown that patients who perceive their offloading device to be heavy have lower offloading adherence [20,61]. This suggests that allowing patients to trial several different knee-high devices and choose the one they perceive to be of lighter weight may lead to higher adherence rates to these gold standard offloading devices [20]. This also suggests that further research and development to reduce the perceived weight and bulk of offloading devices, whilst retaining the important offloading mechanical features and structural integrity of devices that reduce high plantar pressure, is critical to improving future offloading treatments.

### 3.4. Summary of Common Barriers and Solutions

In summary, the best available evidence dispels most perceived barriers of clinicians to prescribing knee-high devices and patients to adhering to use these devices. Thus, there is now a critical need for strategies that will arm clinicians and patients with the evidence-based knowledge that knee-high devices when worn for all weight-bearing activities are much more effective, and typically safer, quicker, and cheaper to use in the real-world to heal most foot ulcers much quicker than any other offloading treatment.

## 4. Future Solutions

There are a number of promising solutions to improve offloading treatments, that have emerged in recent years and that may improve best practice offloading treatments in future. Firstly, improved objective measures that combine plantar pressure, weight-bearing activity, and adherence measures into one overarching plantar tissue stress measure and threshold that potentially predicts time-to-healing at initiation of offloading treatments have shown promise in pilot studies [9,114]. Such a plantar tissue stress threshold could be used in future to enable the development of more personalised offloading treatments for patients, clinicians and researchers, safe in the knowledge that any mix of treatments they choose that keeps their plantar tissue stress levels under a certain threshold should improve their DFU healing [9,114]. Secondly, the ability to continuously measure individual plantar pressure, activity and adherence outcomes, along with DFU healing, have already been incorporated into “smart boots” designed to offload DFU, i.e., knee-high devices incorporated with sensors that measure these outcomes in real time and display them to patients and clinicians on smart watches [9,115,116,117,118]. These smart boots are being armed with alarms to flag the smart watches of patients and clinicians when their plantar pressure, activity or adherence thresholds are exceeded and further treatment is required to improve healing [115,116,117,118]. Thirdly, three-dimensional scanning and printing using smart materials is already available in footwear and closer to also becoming a reality for offloading devices [116,118,119,120]. Such solutions should deliver lighter weight, cooler and less bulky knee-high devices that are more easily customizable to patients wherever they reside. In addition, new smart materials are becoming available that automatically register and redistribute high plantar pressures in real time rather than having to wait to modify devices when patients return to a clinic [20,116,118,119,120]. All these novel solutions are aimed at further reducing high plantar tissue stress, improving patient satisfaction and inturn adherence, and eventually, significantly reducing DFU healing times in future.

## 5. Conclusions

In conclusion, reducing high plantar tissue stress is critical to healing DFU. High-quality evidence demonstrates that knee-high devices, always adhered to, are the gold standard offloading treatment to reduce high plantar tissues stress and more effectively heal DFU. However, studies also show these knee-high devices are somewhat rarely prescribed in clinical practice, and when they are, patients have low acceptance or adherence to using these best practice treatments. There are a number of historical misperceptions of clinicians and patients that are likely to account for this large gap between the high-quality supporting evidence for using knee-high devices and their low use in clinical practice that in the main can be dispelled by best evidence. Thus, more education of clinicians and patients that knee-high devices should nearly always be their first evidence-based choice for offloading treatment, along with further testing of promising novel solutions to improve such devices, seems critical to reducing the high global disease burden caused by DFU in future.

## Figures and Tables

**Figure 1 medicina-57-00941-f001:**
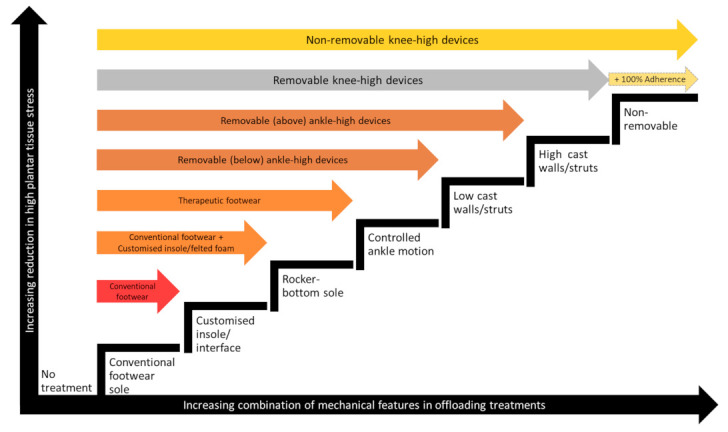
The stairway to best practice offloading treatment for people with diabetic foot ulcers.

**Table 1 medicina-57-00941-t001:** Summary of the effects of offloading treatments on important outcomes, contraindications, and intolerance factors for people with diabetic foot ulcers (DFUs).

Choice	Offloading Treatment	Versus Control Treatment	Healing	Plantar Pressure	Weight-Bearing Activity	Adherence	Patient Satisfaction	Cost- Effectiveness	Adverse Events	Contraindications	Intolerance Factors
**Offloading Devices**											
1st (Gold standard)	Non-removable knee-high device ^	Removable knee-high device	++	=	=	++	=	++	=	Moderate–severe infection	Job requirements
		Removable ankle-high device	++	+	++	++	=	++	=	Moderate–severe ischemia	Frequent driving requirements
		Therapeutic footwear	+++	++	++ *	++	=	+++	=	High falls risk	Unable to attend frequent care
		Conventional footwear	+++ *	+++	+++ *	+ *	-*	+++ *	+ *		Hot climates
1st (Gold standard)	Non-removable knee-high walker	Total contact cast	=	=	=	=	+	+	=	As above Large deformity	As above
2nd	Removable knee-high device	Removable ankle-high device	=	+	+	-	=	=	=	Severe infection	Job requirements
		Therapeutic footwear	++	++	+ *	-*	=	++	=	Severe ischemia	Hot climates
		Conventional footwear	+++ *	+++	++ *	--*	-*	+++ *	+ *	High falls risk	
3rd	Removable ankle-high device	Therapeutic footwear	++	+	+ *	= *	=	++	=	Nil	Nil
		Conventional footwear	+++ *	++	+ *	-*	-*	+++ *	+ *		
**Footwear**											
4th	Therapeutic footwear	Conventional footwear	+ *	+	+ *	--*	-*	+ *	+ *	Nil	Nil
**Surgical offloading**											
Non-healing DFU ^^	Metatarsal head resection, arthroplasty, or osteotomy **	(Non-)removable knee-high device	++	+	+ *	++ *	+	-*	-	Moderate–severe infection	Job requirements
	Achilles lengthening or gastrocnemius recession **	(Non-)removable knee-high device	+	+	+ *	++ *	= *	-*	-	Moderate–severe ischemia	Frequent driving requirements
	Digital flexor tenotomy	(Non-)removable knee-high device	++	+ *	= *	++ *	+	+ *	-	Moderate–severe edema	Unable to attend frequent care

^: Non-removable knee-high device includes customized Total Contact Casts (TCCs) and prefabricated non-removable knee-high walkers (also known as instant Total Contact Casts (iTCCs)); *: Based on expert opinion where no evidence exists; ^^: Non-healing DFU is defined as a DFU that has not reduced in size by >50% after 4–6 weeks of best practice DFU treatment; **: Surgical offloading treatment used in conjunction with (non-)removable knee-high device. Effects, +: small desirable effect (compared to control); ++: moderate desirable effect; +++: large desirable effect; =: similar effect; -: small undesirable effect; --: moderate undesirable effect; ---: large undesirable effect.

## Data Availability

No new data were created or analyzed in this study. Data sharing is not applicable to this article.

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
