# Peer review of "Knee-High Devices Are Gold in Closing the Foot Ulcer Gap: A Review of Offloading Treatments to Heal Diabetic Foot Ulcers"

_medicina, 2021, doi:10.3390/medicina57090941_

Round 1

Reviewer 1 Report

Page 3, lines 112-113 - Please also state the number of patients encompassed by these meta analyses. 

Reviewer 2 Report

A beautiful review widely explaining the state of the art of the offloading in diabetic foot lesions, mirroring the recent guidelines.

Easy to read, clear to understand.

The main question, if the knee-high offloading systems are better than other systems to heal lesions, basing on literature, receive a positive answer, and the conclusions are consistent with the arguments and the bodies of evidences evaluated by authors.
On the other hand, we can not speak of an original aspect, in this review, as the review per se overlaps and reflects equal the guidelines of international societies on diabetic foot.
Nevertheless, the diffusion of such knowledges is still needed and important.
